# The Punishable Child in Sweden—The Tidö Agreement from a Children's Rights Perspective

Jeanette Sundhall [1,*] and Sandra Hillén [2]

1   Department of Cultural Sciences, University of Gothenburg, Box 200, 405 30 Gothenburg, Sweden
2   Faculty of Librarianship, Information, Education and IT, University of Borås, 501 90 Borås, Sweden; sandra.hillen@hb.se
*   Correspondence: jeanette.sundhall@gu.se

**Abstract:** The discourse that has so far dominated in Sweden, and which has manifested itself in various legislation concerning children who commit crimes, is going to change soon. We argue that this discourse is set to be replaced by one that does not consider the subordinate position of children as a result of their age but rather equates them with adults, thus making invisible the power imbalance between children and adults. In this article, we analyze a political document, the Tidö Agreement, and its articulations on youth criminality. We consider the Tidö Agreement to be an important tool in the process of social change, and we carry out this discussion in connection to the United Nations Convention on the Rights of the Child (UNCRC), which became law in Sweden in 2020. By using a discourse theory perspective, we examine the articulations in the Tidö Agreement and discuss how these articulations can reproduce or challenge the current discourses by fixing meaning in certain ways. For instance, the word "child" is ambiguous, and its identity changes depending on how it is positioned in relation to other words in a concrete articulation. In this article, we discuss how this word is used in some contexts but avoided in others, and what consequenses this has.

**Keywords:** children's rights; UNCRC; Tidö Agreement; youth criminality; age order

## 1. Introduction

Organized crime has been a major problem in Sweden for some years, and what is remarkable is that children and young people are a significant part of this problem. The fact that people under the age of 18 are recruited to carry out criminal acts and often they themselves fall victim to gang conflicts makes them both perpetrators and victims in this regard. Explanatory models for how this problem has come to be and how it may be resolved differ depending on who is speaking. To deal with organized crime, the government has rapidly introduced new laws and institutions. The government, which consists of the Moderate Party (M), the Christian Democrats (KDs), and the Liberals (Ls) have, in collaboration with the right-wing nationalist party, The Sweden Democrats (SDs), formulated an agreement in which they have gathered their political ideas for the mandate period 2022–2026, namely, the Tidö Agreement: Contract for Sweden, hereinafter referred to simply as the Tidö Agreement.

In this article, we analyze and discuss the part of the Tidö Agreement that deals with criminality and its possible consequences for children. In Sweden, the tradition of legally treating young people who commit crimes differently compared to adults goes back to the 19th century. It is based on the idea that children and young people who commit crimes are often socially vulnerable and have a special need for support and help and that criminal justice interventions against young people must take the young person's lack of maturity, limited experiences, and special circumstances into account (Holmberg 2022; Proposition. 1997/98: 96 1998). A fundamental idea in Swedish criminal policy is to "try to avoid locking people up, because being locked up has harmful effects on an individual"

(Kriminalvarden.se). A child—in other words, a person under the age of 18—may in principle not be sentenced to prison at all, only if there are special reasons, and until recently, the starting point has been that young people between 18 and 21 should not be punished as harshly as an adult. A life sentence could also not be awarded to someone under the age of 21 (Holmberg 2022; Proposition. 2021/22: 17 2021). The essential view in the Swedish penal system is that people aged 15–17 should be placed in correctional facilities as little as possible, because such placement risks confirming their image of themselves as criminals. The principle is instead that the penalties for young people should primarily be handled by social services. This system of punishment formed quite slowly, with two major reforms taking place in the past 25 years, one in 1999 and one in 2007. Recently, however, the tradition that youth penalties should not be enforced by the correctional service has been broken with the introduction of the penalty of youth supervision on 1 January 2021 and the removal of the so-called penalty discount on 1 January 2022 by the former government (Holmberg 2022). Now, with the new government and the Tidö Agreement, further reforms are coming, not slowly this time, but at speed. In this article, our aim is to investigate how the discourse that has so far dominated in Sweden, and that has manifested itself in various legislation concerning children and young people who commit crimes, is soon to change. Our hypothesis is that the discourse on children being in special need of protection due to their subordinate position in relation to adults, a discourse that, up until recently at least, has dominated Sweden, is being challenged by an opposing discourse, a discourse that equates them with adults. We carry out this discussion in connection to the United Nations Convention on the Rights of the Child (UNCRC), which became law in Sweden in 2020.

## 2. Children as Full Subjects of Rights

According to the UNCRC, children must be seen both as worthy of protection and as holders of their own rights. Children have, through the incorporation of the UNCRC into law in 2020, the right to be full subjects of rights and not merely objects of protection and care. But children depend on adults, and the systems that surround them, to guard their rights. Acknowledging that adults are in a superior position compared to children is one of the starting points of the research field of Childhood Studies (James et al. 1998; Näsman 1994, 2004; Qvortrup 1994). As Elisabet Näsman (2004) discusses, being a "child" is a social position that is determined by relationships to other social positions. The expectations attached to the position are based on these relationships. Childhood and children as social categories are delimited and defined by notions of difference between children and adults. This is manifested in the actuality that we have an age order in society. This age order is distinguished by the fact that age is a widespread and accepted reason for treating people differently. Age and different ways of regulating age are common ways of creating social order (Näsman 2004). That is why age can be regarded as a power order, a social order that carries hierarchies and discrimination, inclusions, and exclusions. Ideas and norms regarding age are used to organize and discipline individuals, activities, and contexts (Krekula et al. 2005; Närvänen 2009). The linear perception of time and the structuring of society based on age have social consequences for the individual. The consequences of being at a certain age vary depending on how age interacts with other social variables such as gender, functional variation, ethnicity, class, and sexuality, but also based on how an individual's age is valued in the historical time and the societal context being studied (Davet and Sundhall 2020). Children are developing, growing up, and maturing, which consists of various biological, psychological, and social processes that they must go through before reaching adult status. In legislation and science, children are largely regarded as individuals in the future, and childhood is thus seen as a period of lack (Näsman 2004). In this article, we discuss, among other things, age limits. Age limits give an indication of how society considers children to be different from adults, and childhood contains several age limits, unlike adulthood. Some age limits exist to protect children, others to accommodate children's rights with reference to their needs. Childhood, compared to other life phases, consists of an intricate age regulation that refers to several widely different areas of life,

relationships, and conditions (Näsman 2004, p. 59). The current age at which a person is considered capable of distinguishing between right and wrong and therefore can be held legally responsible for a crime is 15 years in Sweden. The day you turn 15, you are also allowed to drive a moped, watch movies prohibited for children, ride a bike without wearing a helmet, transport a person on a bike or moped if that person is under 10 years of age, and have sex. However, the legal measures for children over 15 involved in crime are different from the measures for those over 18 years of age and up until now have included, in place of imprisonment, special sanctions such as youth care, youth service, youth supervision, and closed youth care.

### 3. The Tidö Agreement—A Crucial Document

The document analyzed and discussed in this article, the Tidö Agreement, received a lot of attention when it was presented to the public just over a month after the parliamentary elections were held on 22 September 2022. The document is named after the 17th-century castle Tidö, where the agreement was made and written down over the space of two days in October 2022. It is a simply designed document consisting of 62 pages. The style is concise and austere. Seven directives of collaborative projects constitute the content: Health and Medical Care, Climate and Energy, Criminality, Migration and Integration, School, Growth and Household Economy, and Other Collaboration Issues. In the introduction, the four parties formulate the aim of the agreement: "The collaboration will lay the foundation for a long-term sustainable collaboration, with the aim of implementing reforms that solve the major social problems Sweden has..." (Tidöavtalet 2022, p. 2). Although formulated as proposals, the intention is for these proposals to be implemented. The Tidö parties, especially the Sweden Democrats, have been very clear about this (Pelling 2023, p. 7). In this article, our analysis focuses on the Directive collaborative project Criminality. In another article, we focus on the Directive collaborative project Migration and Integration. In the present article, we analyze the Tidö Agreement with a discourse theory approach (Laclau and Mouffe 2001). An important starting point in discourse theory is that social phenomena are never complete or total; meaning can never be definitively fixed. This means that there is always room for an ongoing social battle over definitions of society and identities—and the outcome of that battle has social consequences. We consider the Tidö Agreement to be a crucial document in terms of its intention of (re)shaping Swedish society and as an important tool in a process of social change.

The analysis method is closely linked to the discourse theory's starting point of the investigation. Discourse theory emphasizes how it is crucial to focus on concrete expressions, the *articulations* (Laclau and Mouffe 2001). "Child", for instance, is an ambiguous word, or sign, and its identity changes as it is used in relation to other words in a concrete articulation. Articulation is thus a concept that captures both change and reproduction. An articulation can reproduce or challenge current discourses by fixing meaning in certain ways. Each expression is an active reduction in meaning possibilities because each expression puts signs in particular relations to each other (Laclau and Mouffe 2001; Winther Jørgensen and Phillips 2000, p. 35). For us, it is crucial to examine which discourse a specific articulation is based on and which discourses the articulation is reproducing. As university employees in Sweden, it is part of our mission to promote democratic values (forvaltningskultur.se n.d. accessed on 19 December 2023), and we are deeply concerned about the rapid changes in the Swedish constitution that the Tidö Agreement suggests and the consequences for children and young people living here.

### 4. Results

The overall section that we focus on in this article is the Directive collaborative project Criminality, which constitutes 10.5 pages of the 62-page long Tidö Agreement. The expressed purpose of the collaborative project is as follows:

> ... to develop and implement concrete political proposals that solve Sweden's most important societal problems regarding crime and gang crime with the goal of

increasing security, preventing younger people from being drawn into crime, ensuring that more crimes are investigated and lead to prosecution, and, to combat serious organized crime, also ensuring that crime victims receive increased compensation and that fair punishments are given to criminals. (Tidöavtalet 2022, p. 18)

For the reader to gain an understanding of the kinds of proposals in the Tidö Agreement and how they are articulated, we will discuss its various headings, subheadings, and middle headings and point out what we find particularly problematic in them when it comes to children's rights.

Under the heading Reforms that will be implemented in the project, the first subheading is named Pattern-breaking measures to stop the gangs with the following middle headings: Secret means of coercion, Expulsion of security threats, Double punishment for gang criminals, Visitation zones, Anonymous witnesses, Criminalization of participation in criminal gangs, Prohibition of stay, New main rule in privacy legislation, Mandatory detention in more cases, and Forfeiture (Tidöavtalet 2022, p. 18ff). Before we start discussing the part on criminality that is particularly directed toward children and youth, we will discuss the very first paragraph in the overall Directive collaborative project Criminality, which, we believe, sets the tone for how the rest of the text should be read. Under the heading "Reforms to be implemented in the project—Expulsion of security threats" (Tidöavtalet 2022, pp. 19–20), the following suggestion is formulated:

Deport more gang criminals. The possibility of deporting gang criminals who lack Swedish citizenship without them having been convicted of a crime must be investigated. Such an opportunity exists today for other system-threatening crimes through the Act (2022: 700) on control of certain foreigners. Persons who can be linked to organized crime must be added to the prerequisites for deportation, according to this law. Furthermore, the number of foreign citizens who are active gang criminals should be calculated. (Tidöavtalet 2022, pp. 19–20)

In this articulation, "gang criminals" is explicitly put in relation to non-Swedes, that is, people who lack Swedish citizenship. This concerns persons referred to as "gang criminals" and also "persons who can be linked to organized crime", which are suggested to be deported without having been convicted of a crime. The articulation "linked to organized crime" is a vague and imprecise expression that risks having serious consequences for persons who are categorized as fitting into this description. The last sentence, "Furthermore, the number of foreign citizens who are active gang criminals should be calculated", puts "gang criminals" and non-Swedes in direct relation to each other. These articulations build upon and reproduce the well-known discourse on the criminal immigrant, and they exclude connections to other kinds of criminal networks or criminal gangs, such as biker gangs. In this way, the Tidö Agreement aims to establish and strengthen an unambiguous link between "the other", the foreigner, and criminal behavior. Moreover, it is arguable that this link is facilitating the changes that are currently being made in Swedish migration policy.

The second subheading under the heading *Reforms that will be implemented in the project* can roughly be translated as *Action (krafttag) against juvenile delinquency.* There is no synonym in English for the Swedish word "krafttag", but it can be regarded as a combination of the words *force* and *hold*. This articulation, with the use of the word *krafttag*, is distinctive for the short-cut and harsh style of the Tidö Agreement, as the expression conveys action and determination. Under this subheading, there are the following middle headings: *Responsibility for young people who are serious criminals, Punishment for young offenders, New penalty for young people, 24 h guarantee in social services, The Act on Care of Young People, The Young Offenders Act*, and *Parental responsibility* (Tidöavtalet 2022, p. 21f). In these middle headings, the word "young" is used frequently. It is a practical word in the context, as it covers a wide age range, but it is not unproblematic from a discourse analysis standpoint, where changes in articulations have certain consequences and imply a reduction in the possibilities of meaning, as we will later discuss further.

Now, we will discuss some of the proposals in this section directed toward youth criminality, beginning with the middle heading *Punishment for young offenders*:

> The penalty reduction for those over 18 years of age must be removed. An investigation must review the penalty discount for those under 18 and at the same time consider lowering the age of criminal jurisdiction. (Tidöavtalet 2022, p. 22)

Here, age is articulated in various ways. In the first sentence, the articulation is "those over 18 years of age", which refers to the age categorization of adults but without mentioning the word adult. In the next sentence, the articulation is "those under 18", which refers to the age categorization of children but without mentioning the word child. The designation of the age groups child and adult is avoided by choosing articulations that revolve around chronological age, in this case, the age of 18, which is the age of maturity and the age limit that legally separates children from adults. The choice to use synonyms for age categories, like every choice, has consequences, and the frequency with which this choice is made highlights how it is a deliberate one.

Under the middle heading *Responsibility for young people who are serious criminals,* a proposal on youth prisons is presented:

> Furthermore, special youth prisons are to be set up, for which the Correctional Service is to be the principal. Youth prisons are to replace the special youth homes that SiS is responsible for today, where sentences to closed youth care are normally enforced. The maximum time for closed youth care must also be extended, and anyone who turns 18 while serving a sentence in a youth prison must be transferred to a regular institution. (Tidöavtalet 2022, p. 21)

The proposal on youth prisons received a lot of attention when the Tidö Agreement was presented, as did the previous proposal on lowering the age of criminal responsibility. The articulation in the proposal does not mention the word *child* in relation to the word *prison*, even if it should be a relevant choice, since every person under the age of 18 is to be considered a child. "Anyone who turns 18 while serving a sentence in a youth prison" has the same meaning as *A child who becomes an adult while serving a sentence in a child prison.* Something else is also done here: The word *youth* is put in relation to the word *prison*, and the words *home* and *care* are suggested to be removed from *youth*. An articulation can reproduce or challenge current discourses by fixating meaning in certain ways. Home and care belong to today's discourse around *youth*, but the above proposal suggests a future that will look different. The discourse that children should be granted special protection because of their age is about to be replaced by a discourse that does not consider the subordinate position of children but equates them with adults, thus making invisible the power imbalance between children and adults. Under this middle heading, *Responsibility for young people who are serious criminals,* the term *youth* occurs seven times, *youth prisons* three times, *youth homes* and *youth care* once each, and the term *child* once. The one time that a *child* occurs, the word *parents* is also present.

> An obligation for social services must be introduced to call a child's parents to a meeting within 24 h after a child has been arrested for a crime so that social services can get the parents to participate in efforts to support the young person.
> (Tidöavtalet 2022, p. 21)

In this articulation, where parental responsibility is invoked, it seems unproblematic to use the word child. As discussed, *child* is an ambiguous word, open to shifting meanings, and its identity changes when put in relation to other words in a concrete articulation. A place where *children* are used frequently—six times in fact—is in the middle heading *Parental responsibility*.

> An investigation must review various possibilities to strengthen parental responsibility and enable early intervention for children who commit crimes, are at risk of doing so, or live in other forms of vulnerability. This involves, among other things, reviewing the possibilities for additional support for children.

The investment in parental support programs is expanded with the goal that it should be available in all municipalities in the country and that investment in leisure cards is implemented. Furthermore, the Social Services Act must be amended with the aim of giving social services extended powers to decide on early and mandatory interventions for children or their guardians, that is, so-called intermediate coercion. In all placements of children, increased efforts must be made to ensure functioning schooling. Economic and social consequences can have an impact on how parents exercise their parental responsibility over children who are at risk of getting involved in crime. In the past, there have been certain opportunities to impose joint and several liability for damages for children's delinquency on guardians. Other ways of influencing parents through economic and social consequences should be tried. It may be about changes in the custodian's liability for damages or other financial measures. (Tidöavtalet 2022, p. 23)

In this proposal, there are investments that will benefit children, such as parental support programs and additional support for children. However, there are also suggestions on economic and social consequences for parents. In this proposal, it is necessary to use the word *child/children*. If the person involved in a crime is over 18 years of legal age, no parental responsibility can be claimed. Here, the Tidö Agreement has no choice but to be clear that it is children who are referred to, not the more imprecise concepts of "youth" or "young person".

Another example of when the Tidö Agreement uses words that have to do with age is to be found under the middle heading *New penalty for young people*:

A new penalty, extended youth supervision, is to be introduced as a penalty for young people, with an expanded toolbox for the police to body search the young person, be able to search the young person's house after a prosecutor's decision, and also be able to obtain court permission for covert coercive measures. (Tidöavtalet 2022, p. 22)

This is not noticeable in the English translation, but in Swedish, there is a difference between the words *unga/young person* in the feminine form that ends with an -a, which is usually considered the basic form of a word, and the word in the masculine form, which ends with an e: *unge/young person.* In the Tidö Agreement, the feminine/basic form of the word is used almost everywhere (34 times, in fact), while the masculine form is used in just 2 exceptional cases. In the proposal above, it is in this sentence: *body search the young person.* Here, it seems necessary to point out that the young person who is going to be body searched is not a feminine person and certainly not a child.

In the third subheading, *A complete and comprehensive review of the criminal legislation is carried out*, there are not many suggestions that are explicitly directed to younger people, even though several of them certainly will have an impact on young people's lives. One middle heading, *Revised regulations regarding statutes of limitations*, however, addresses the issue of young people with this articulation:

Special juvenile sanctions for young people who evade justice and have reached the age of adulthood must be converted into equivalent penalties for adults before the penalty is enforced. (Tidöavtalet 2022, p. 24)

This is interesting from an age-limit perspective since a crime committed by a child this way will automatically be punished according to an adult punishment scale when the child turns 18.

On the other hand, the paragraph continues to state the following:

Furthermore, the statute of limitation shall be eliminated for all forms of sexual offenses against children under the age of 18. (Tidöavtalet 2022, p. 24)

Here, it becomes clear that young people in this sentence are defined as victims (of sexual offense) instead of perpetrators and are then called *children under the age of 18.* This

strengthens our thesis that the word *child/children* is only used when it is unproblematic and when children's need for protection is emphasized.

Under the fourth subheading, *Other reforms,* one middle heading that is directed toward young people is *Try a system of juvenile delinquency boards and allow evidentiary proceedings against young people in more cases*, and here, the inspiration is taken from Sweden's neighboring country, Denmark:

> In Denmark, there is a system of juvenile delinquency boards led by a judge, with representatives from the police and the municipality. The aim is to take earlier, more consistent, and powerful action against the background of the development of gang environments where criminals target vulnerable and impressionable children and young people. An investigation into introducing a similar system is underway. However, the assignment must not be reported until 30 August 2024 and should be completed earlier. Evidentiary suits (bevistalan) against children under the age of 15 must be used in more cases. (Tidöavtalet 2022, p. 27)

Here, the word *children* is used in the formulation "children under the age of 15". This is, of course, a tautology since all persons under the age of 15 are children, and it would be enough to formulate the sentence "persons under the age of 15" for the meaning to come across. A significant aspect of referring to "children under the age of 15" is that these persons have not reached the age of criminal responsibility, according to the legislation in Sweden when the Tidö Agreement was formulated. The term evidentiary suit means that you do indeed investigate whether a crime has been committed, that is, investigate the issue of guilt, and this is only done when the prosecutor judges that it is possible to prove that the person committed the crime. So, the court takes a position on whether the child committed the crime or not but will not impose any penalty (aklagare.se n.d.). We wonder what the point of such an approach is and why it is outlined in the Tidö Agreement. In the same paragraph, children and youth are referred to as "vulnerable and impressionable children and young people", which is a rare formulation in the Tidö Agreement. In this articulation, children and young people are referred to in relation to "criminals". Here is an articulation that makes the unequal power relations in the age order visible—an iniquity that is otherwise rarely visible in the Tidö Agreement. The distinction between "children and young" people on the one hand and "criminals" on the other hand suggests the driving forces behind organized crime: adults.

The Tidö Agreement also contains several proposals that affect children's legal status and suggestions that limit children's freedom of movement, integrity, and privacy, even though children or young people are not mentioned in these contexts at all, for instance, suggestions regarding *Secret means of coercion*, *Visitation zones,* and *Mandatory detention in more cases*. The consequence of these articulations is a replacement of the discourse on children being an age categorization with certain rights due to their low age with a discourse where there are no differences between children and adults. The consensus is that when a crime has been committed, children should be treated the same as adults.

## 5. Discussion

Our analysis of the Tidö Agreement is based on how children's rights are expressed in the UN Convention on the Rights of the Child. We have shown how in the *Directive collaborative project Criminality*, a series of proposals are put forward that deal with the possibility of increasing coercive measures and being able to impose harsher punishments on those lower in age, with the aim to "prevent younger people from being drawn into crime" (Tidöavtalet 2022, p. 18).

Several proposals from the Tidö Agreement are now being implemented. Visitation zones, which the government is now calling safety zones, are proposed to be implemented at the time of writing. An intervention in the zone may take place without there being any suspicion of a crime. The possibility of conducting visitation on children is also pronounced in this statement. Critics of this have pointed out a risk of arrest due to discriminatory ethnic profiling, partly because the zones are connected to geographically bounded areas

(Tanaka 2024). We consider it a significant risk that these changes will affect some children, in particular, children who, already today, are reporting to children's rights organizations on being subjected to visitations and identity checks because of how they look and where they live (Rädda Barnen 2022, p. 5). Safety zones violate children's right to integrity, privacy, and freedom of movement, as articulated in Article 16 of the UNCRC. We have shown how some articulations build upon and reproduce the well-known discourse on the criminal immigrant. In relation to this, it is important to emphasize that the UNCRC applies to every child without discrimination, as stated in Article 2 (Ohchr.org n.d.). The Committee on the Rights of the Child particularly underlines discrimination that can be "the result of a lack of a consistent policy and involve vulnerable groups of children" (UN Committee on the Rights of the Child 2007, p. 4). The Committee considers the following groups of children as vulnerable: "street children, children belonging to racial, ethnic, religious or linguistic minorities, indigenous children, girl children, children with disabilities and children who are repeatedly in conflict with law" (UN Committee on the Rights of the Child 2007, p. 4).

The proposal on lowering the age of criminal responsibility has received particular attention from the UN Committee on the Rights of the Child (2023), the members of which have written a general comment regarding this, stating that they are "deeply concerned about current moves to lower the minimum age of criminal responsibility" and that they urge Sweden "to maintain the minimum age of criminal responsibility at 15 years of age" (UN Committee on the Rights of the Child 2023, p. 13). Regarding the proposal for youth prisons, the government has given the Correctional Service the task of planning for the institutions. The idea of youth prisons is a very new one in Sweden. Until now, children may, in principle, not be sentenced to prison at all. However, the discourse in this regard is about to change. Youth prisons are a clear violation of the UNCRC, which in Article 37 states that "(n)either capital punishment nor life imprisonment without possibility of release shall be imposed for offences committed by persons below eighteen years of age" (Ohchr.org n.d.). We have shown how the words "young" and "youth" are frequently used instead of words such as "children" or "minors". We have also pointed out that "child" is used in relation to certain areas, for instance, in relation to parenthood, but is avoided in relation to words commonly associated with adults, for instance, prisons. Only on one occasion is the age order present in the Tidö Agreement, in the articulation on how "criminals target vulnerable and impressionable children and young people".

Not every suggestion for changes in the field of criminality in the Tidö Agreement counteracts children's rights, though; some proposals are intended to strengthen these rights, such as propositions that aim to reinforce the position of crime victims and the possibility to investigate whether young offenders have been influenced by older persons. This responds to a newly enforced law that makes it illegal to use and recruit adolescents for criminal activities. An inquiry is suggested that will investigate various possibilities to strengthen parental responsibility and enable early interventions for children who commit crimes. Social services are expected to take on an extended role linked to more obligations regarding preventing juvenile delinquency. Together, these suggestions show a shift in which a greater responsibility for preventing young people's criminality is placed on parents and social services than before. The Committee on the Rights of the Child indicates in General Comment No. 10 (UN Committee on the Rights of the Child 2007), Children's Rights in Juvenile Justice, that Articles 18 and 27 confirm the importance of the responsibility of parents for the upbringing but reminds that the CRC also requires state parties to provide the necessary assistance to parents and other caregivers. The Committee underlines that this should not only focus on the prevention of negative situations but "also and even more on the promotion of the social potential of parents" (UN Committee on the Rights of the Child 2007, p. 8). The Committee states that it "...regrets the trend in some countries to introduce the punishment of parents for offences committed by their children" (UN Committee on the Rights of the Child 2007, p. 8). The Committee believes that this could be appropriate in some cases but underlines that they do not believe that criminalizing parents will contribute to them becoming active in the reintegration of their

child (UN Committee on the Rights of the Child 2007, p. 16). There are, however, certain proposals in the Tidö Agreement where it seems that the ambition is to strengthen children's rights but where children's rights organizations have warned that the opposite could result (Rädda Barnen 2022). The overall picture of the *Directive collaborative project Criminality* is an indication that the age limits that exist for the purpose of protecting children, such as the age of criminal responsibility and the legal conditions for children not to be put in prison in Sweden, are renegotiated. Although the directive contains proposals intended to protect children from being recruited and exploited in organized crime, the solution is more often discussed based on legal ideas where harsher punishments, shifted age limits, and limited mobility are prized over preventive solutions and a children's rights perspective. Children's rights organizations in Sweden have pointed out that some proposals in the Tidö Agreement do strengthen children's rights to some extent, but that these parts do not outweigh the negative effects of the proposals that weaken children's rights (Rädda Barnen 2022). There are proposals that violate children's rights directly or in their practical application. In relation to the state, citizens are both objects and subjects regardless of age. What separates children from adults is not a completely different status but the extent of their own room for action and their own responsibility within all these areas of rights (Näsman 2004). Children's rights are primarily based on adults complying with Article 3 and putting the best interests of the child first in all actions concerning children. When children's rights are discussed, it is of great relevance and importance to point out children's difference compared to the adult subject. This difference is, amongst other things, about children being dependent on adults. The UN Committee on the Rights of the Child (2007) describes this difference in Article 3: "Children differ from adults in their physical and psychological development, and their emotional and educational needs. Such differences constitute the basis for the lesser culpability of children in conflict with the law. These and other differences are the reason for a separate juvenile justice system and require a different treatment for children. The protection of the best interest of the child means, for instance, that the traditional objectives of criminal justice, such as repression/retribution, must give way to rehabilitation and restorative justice objectives in dealing with child offenders" (UN Committee on the Rights of the Child 2007, p. 5).

Alan Prout stated that "(m)odernity constituted childhood as the 'cultural other' of adulthood" (Prout 2005). This condition is incessantly constituted. The UNCRC is characterized by the awareness of how childhood is constituted and the fundamental ideas about children having complete human dignity while at the same time being in need of special support and protection, and it is adults who must ensure that children's rights are maintained and achieved. Article 40 states that "…every child alleged as, accused of, or recognized as having infringed the penal law to be treated in a manner consistent with the promotion of the child's sense of dignity and worth, which reinforces the child's respect for the human rights and fundamental freedom of others and which takes into account the child's age and the desirability of promoting the child's reintegration and the child's assuming a constructive role in society" (Ohchr.org n.d.). The Committee states that "this principle must be applied, observed and respected throughout the entire process of dealing with the child, from the first contact with law enforcement agencies all the way to the implementation of all measures for dealing with the child" (UN Committee on the Rights of the Child 2007, p. 6) and that this demands that all professionals, that is, adults dealing with this child, must have knowledge about child development, how to achieve children's well-being and also about "the pervasive forms of violence against children"(UN Committee on the Rights of the Child 2007, p. 6). The need for a child impact analysis is also not stated anywhere in the part of the *Directive collaborative project Criminality*, despite the fact that the directive will affect the lives of many children. Article 12 on the right to be heard should of course also be considered in the case of juvenile justice and in every stage of the process. The child's right to express themselves is one of the portal principles, and the Committee indicates in General Comment 10 (UN Committee on the Rights of the Child 2007) that "the voices of children involved in the juvenile justice system are increasingly

becoming a powerful force for improvements and reform, and for the fulfilment of their rights" (UN Committee on the Rights of the Child 2007, pp. 5–6). We do have an ongoing dark situation in Sweden, with underage children being viewed as, and used as, collateral damage in organized crime. Although the suggestions made in the Tidö Agreement are meant to be the solution to this situation, our analysis shows that there are bigger things at stake and many questions to be asked. More repressive methods and tools may seem like a quick fix, but will more surveillance that targets some children based on their appearance and where they live increase their trust in society? And if children are more malleable and vulnerable, should they spend their adolescence in prison with adult perpetrators? Since children are the center of attention in the debate of lethal violence and organized crimes, where are all the progressive thoughts about how to prevent them from seeing gangs as sources of opportunity? It is also important to point out that the ideas of the Tidö Agreement are realized in a time of major cutbacks in preschools, schools, health, and social care that will impact children's lives in many ways. Instead of focusing on prevention and children's equal opportunities that will prepare children for their future, the Tidö Agreement focuses on punishing children for the massive failure of the adult world. This, and the rapidness with which these fundamental changes have been enforced in a country that incorporated the UNCRC into its legislation less than four years ago, is why the Tidö Agreement should be regarded as a great betrayal on behalf of children.

**Author Contributions:** Conceptualization, J.S. and S.H.; methodology: J.S.; formal analysis J.S. and S.H.; writing—original draft preparation, J.S.; writing—review and editing, J.S. and S.H. All authors have read and agreed to the published version of the manuscript.

**Funding:** This work was funded by the Swedish Research Council under Grant 2020-03888.

**Institutional Review Board Statement:** Not applicable.

**Informed Consent Statement:** Not applicable.

**Data Availability Statement:** No new data were created or analyzed in this study. Data sharing is not applicable to this article.

**Conflicts of Interest:** The authors declare no conflicts of interest.

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
