# Peer review of "The Punishable Child in Sweden—The Tidö Agreement from a Children’s Rights Perspective"

_socsci, doi:10.3390/socsci13040215_

Round 1

Reviewer 1 Report (New Reviewer)

Comments and Suggestions for Authors

I congratulate the authors on dealing with a highly political and controversial contemporary subject, with a sound and ethical analysis that challenges adult-centric legal reforms. 

I encourage the authors to also connect with more recent developments in childhood studies e.g., the movement towards childism. The authors rely on theoretical insights from childhood studies and cite central works, which have nevertheless been developed further. Given that the author's hint towards a larger social critique of discipline and punishment mechanisms, they might connect with more recent developments in their conclusion. 

This suggestion is optional, and the author's might consider it if they wish.

Comments on the Quality of English Language

The language is clear and precise. A final proof-reading before submission is recommended.

Author Response

Dear reviewer 1,

We are very happy that you find our work interesting. To respond to your point that childism could be a concept to take in, we would like to point out that our focus is on the discursive changes that the agreement entails, and its consequences, rather than ethical conciderations of what a child is. According to the UN Convention of the right of the child, children are labeled as both humans with legal rights, as well as a group in need of protection. 

Childism, in the meaning of children as equal worthy as adults are not that radical in that sense and in the context of the convention. Childism in the context of dealing with prejudice and/or discrimination against the young or as a way of empowering children as an oppressed group could have been one possible angle, but in this case, we try to show this through our analysis of the language used in the Tidö-agremeent. 

Reviewer 2 Report (New Reviewer)

Comments and Suggestions for Authors

I love the potential of this article and applaud the authers for bringing their brilliant rigor to conducting a deep analysis of the Tido agreement from a child rights perspective. My only criticism--and it is significant--is that they have not deeply engaged with the U.N. Convention on the Rights of the Child despite repeatedly referencing it. I think this can be accomplished in in 6 to 8 additional hours of research and writing so it should not be a heavy lift. Specifically, I would encourage the authors to look at Gen. Comment 10 of the UN Committee on the Rights of the Child, which will provide them a very succinct road map. I recommend that this article is accepted on the condition that it is further developed in that regard. Otherwise, it looks great!

Author Response

Dear reviewer 2,

We are very happy that you appreciate our work and we have taken your comments, which we found very helpful, into account. In order to respond to your point that the discussion of the Convention on the Rights of the Child needed to be deepened, we have included the discussion of the Committee on the Rights of the child, comment number 10. We have deepened the discussion about how the  changes may effect children and particularly vulnerable children in line with point made of the Committee on the Rights of the child.

Round 2

Reviewer 2 Report (New Reviewer)

Comments and Suggestions for Authors

Thank you so much for expanding your discussion of the CRC in relation to the Tido Agreement. It is much more balanced now and I look forward to seeing your article "in print."

This manuscript is a resubmission of an earlier submission. The following is a list of the peer review reports and author responses from that submission.

Round 1

Reviewer 1 Report

Comments and Suggestions for Authors

Please see attached. This would make a wonderful chapter in a book. 

Comments on the Quality of English Language

Recommend English language proofing service.

Author Response

Reviewer 1

Reviewer: Line 22 - Author seems to equate organized crime with gang crime. These are generally thought of as two separate categories of crime. Author should define what they mean by “organized crime” at the beginning of the paper.

Response: We do not equate organized crime with gang crime. We are discussing precisely those concepts used in the societal debate, as it is being conducted in Sweden right now. We are problematizing the concept gang  crime as part of our discourse analysis.

Reviewer: Lines 43-67 – There are a lot of different ideas here, some of which may not be entirely relevant for the paper. It seems like the authors should focus on recruitment of youth for organized crime ends and eliminate the etiology of why youth commit crime. That seems like a separate paper.

Response: This is solved since we rewrote the article

Reviewer: Line 65 – Authors note that Barnen has examined how the agreement may affect children. Since that is the main thrust of this paper, that article should be discussed in-depth here. The gaps must be pointed out, or some indication of how the author’s paper will add to what is already known. The current paper is not adequately set up as is with the current literature review.

Response: Barnen = Rädda Barnen = Save the Children. It is a report, not an article, and since we rewrote the article, this is solved. The last sentence is a bit unclear but if the reviewer is asking for a literature review, we do not think it is relevant for our article, since our main focus is on the Tidö Agreement, a brand new document where no research has been done so far.

Reviewer Lines 69-78 Introduction to the Agreement should have been made in the Introduction portion of the paper. The key sections on the UN Rights of the Child also should have been included at the beginning.

Response: We believe that our argumentation is clear as it is now since we rewrote the entire aricle.

Reviewer Line 80 – The discourse analytical approach utilizes theory, yet no theory is presented prior to this to set up the analysis. A theory discussion should be placed before this section.

Response: We have now a more developed discussion on method- which includes theory, both in the Material and Method-section and we show how we are using the method in our analysis.

Reviewer Line 87 and paragraph – Much of this is background information that should be at the beginning of the paper in order to set up the research aim to explore how the Agreement aligns with the UN Rights of Children.

Response This is solved since we rewrote the article

Reviewer Line 161 – Paradigms need to be articulated earlier and that may be the best way to frame the article at the beginning – show a paradigm shift from rehabilitation/nurturing to classical criminological approach.

Response: One part of our rewriting is that we now focus on to show how the current discourse is challenged by a new one, and we believe that this is made clear in our analysis.

Reviewer: Results & Discussion

There are some good ideas here, but they seem to be “forced” into these headings, which should not be combined (results are separate from discussion). It would be more fruitful for the authors to write this as an analysis for a chapter, thus freeing them to use different headings that are more aligned with their paper.

Response: We have now separate sections for results and discussion and we believe that our argumentation is much more solid now.

Reviewer: Line 370 and corresponding paragraph – this seems to be a quote, and thus should be designated as one. The paragraph needs to be integrated within the conclusion.

The conclusions are commentary in nature, and while the authors make good points, this is a

commentary, not rooted in original social science research disseminating new knowledge.

Response: The quote is deleted  and we do not have a conclusion-section anymore. However, we believe that the points we make now, in the Discussion, is very relevant and also founded in our analysis, in a completely different way than in the previous version of the article.

Reviewer 2 Report

Comments and Suggestions for Authors

This is a review and analysis of a  62 page document/agreement produced in Sweden on future governance for future handling of child crimes given an expansion of such crimes by organized crime, especially in light of gang violence.

The article mainly uses United Nations documents and suggestions as a point of critique of measures to raise punitiveness and juvenile responsibility standards for crime. It purports to use critical analysis but beyond use of these materials there really is not much depth to the critique. It reads like a political editorial and I would go further to say that it is one.

There is one social linguistic point that is made about reluctance to use the words child and children when advocating toward enhanced punitiveness.  But, that is the only linguistic analysis in the paper so it seems a little odd.

The review of literature on the topic is thin as is analysis of the document. It amounts to a disagreement that the authors hold with the content of the document, and in my opinion that does not merit publication in a journal about the social sciences. 

The grounding of a critique of national policy in the guidance from international elites at the United Nations seems to not be very deep critical analyses.  Yes, different political institutions have differences but the one that the authors agree with, the international perspective from the U.N.,  doesn't necessarily offer an objective reality or a standard to match against a 

Comments on the Quality of English Language

The English use is solid.  I saw some minor things.

The use of "this month" in an academic article with no time given.

themselves on line 22.

A stray sentence about explanatory models on line 23.

The question with the word noticeable on line 40.

Punctuation between "children. That" on line 88.

Use of the word lack on line 109.

"legal responsible" on age 121. 

Initiatives that deal on line 290.

conditions and conditions line 290.

Author Response

Reviewer 2

R2: The article mainly uses United Nations documents and suggestions as a point of critique of measures to raise punitiveness and juvenile responsibility standards for crime. It purports to use critical analysis but beyond use of these materials there really is not much depth to the critique. It reads like a political editorial and I would go further to say that it is one.

Response: Now, since we rewrote the article, and are using a more specific kind of discourse analysis, we believe that there really is depth to the critique in a complete different way than in the previous version of the article.

R2: There is one social linguistic point that is made about reluctance to use the words child and children when advocating toward enhanced punitiveness. But, that is the only linguistic analysis in the paper so it seems a little odd.

Response: Now, we have plenty of examples of how this is being done.

R2: The review of literature on the topic is thin as is analysis of the document. It amounts to a

disagreement that the authors hold with the content of the document, and in my opinion that does not merit publication in a journal about the social sciences.

Response: The review of literature: There are research that in some way connects to our topic,  about the criminal age of responsibility in other countries for instance. But our aim is not to discuss these matters as such. Our aim is to discuss how, in the Tidö Agreement, which we consider to be an  important tool in a process of social change,  the current discourse is about to be replaced. In that sense, we have a meta perspective, where our focus is how this is done.

The analysis is now much more detailed and exhaustive. As we still disagree with the content of the document, we are now, thanks to the method,  able to show exactly why it is so problematic.

R2: The grounding of a critique of national policy in the guidance from international elites at the United Nations seems to not be very deep critical analyses. Yes, different political institutions have differences but the one that the authors agree with, the international perspective from the U.N.,doesn't necessarily offer an objective reality or a standard to match against a negotiated plan from national elites.

Response: We believe that the situation regarding children’s rights is very serious in Sweden right now. As we write in the article: the ideas of the Tidö Agreement are realized in a time of major cutbacks in preschools, schools, health, and social care that will impact children’s lives in many ways. Instead of focusing on prevention and children’s equal opportunities that will prepare children for their future, the Tidö Agreement focuses on punishing children for the massive failure of the adult world. This, and the rapidness with which these fundamental changes have been enforced in a country that incorporated the UNCRC into its legislation less than four years ago, is why the Tidö Agreement should be regarded as a great betrayal on behalf of the children.And because of this comment, we felt obliged to put this sentence in the article: As University employees in Sweden, it is part of our mission to promote democratic values (forvaltningskultur.se/statliga-vardegrunden/) and we are deeply concerned about the rapid changes in the Swedish constitution that the Tidö Agreement suggests and the consequences for children and young people living here.